# Learn What Not to Learn: Action Elimination with Deep Reinforcement Learning

**Tom Zahavy**[*1,2], **Matan Haroush**[*1], **Nadav Merlis**[*1], **Daniel J. Mankowitz**[3], **Shie Mannor**[1]

[1]The Technion - Israel Institute of Technology, [2] Google research, [3] Deepmind

Corresponding to {tomzahavy,matan.h,merlis}@campus.technion.ac.il

## Abstract

Learning how to act when there are many available actions in each state is a challenging task for Reinforcement Learning (RL) agents, especially when many of the actions are redundant or irrelevant. In such cases, it is sometimes easier to learn which actions **not** to take. In this work, we propose the Action-Elimination Deep Q-Network (AE-DQN) architecture that combines a Deep RL algorithm with an Action Elimination Network (AEN) that eliminates sub-optimal actions. The AEN is trained to predict invalid actions, supervised by an external elimination signal provided by the environment. Simulations demonstrate a considerable speedup and added robustness over vanilla DQN in text-based games with over a thousand discrete actions.

## 1    Introduction

Learning control policies for sequential decision-making tasks where both the state space and the action space are vast is critical when applying Reinforcement Learning (RL) to real-world problems. This is because there is an exponential growth of computational requirements as the problem size increases, known as the curse of dimensionality (Bertsekas and Tsitsiklis, 1995). Deep RL (DRL) tackles the curse of dimensionality due to large state spaces by utilizing a Deep Neural Network (DNN) to approximate the value function and/or the policy. This enables the agent to generalize across states without domain-specific knowledge (Tesauro, 1995; Mnih et al., 2015).

Despite the great success of DRL methods, deploying them in real-world applications is still limited. One of the main challenges towards that goal is dealing with large action spaces, especially when many of the actions are redundant or irrelevant (for many states). While humans can usually detect the subset of feasible actions in a given situation from the context, RL agents may attempt irrelevant actions or actions that are inferior, thus wasting computation time. Control systems for large industrial processes like power grids (Wen, O'Neill, and Maei, 2015; Glavic, Fonteneau, and Ernst, 2017; Dalal, Gilboa, and Mannor, 2016) and traffic control (Mannion, Duggan, and Howley, 2016; Van der Pol and Oliehoek, 2016) may have millions of possible actions that can be applied at every time step. Other domains utilize natural language to represent the actions. These action spaces are typically composed of all possible sequences of words from a fixed size dictionary resulting in considerably large action spaces. Common examples of systems that use this action space representation include conversational agents such as personal assistants (Dhingra et al., 2016; Li et al., 2017; Su et al., 2016; Lipton et al., 2016b; Liu et al., 2017; Zhao and Eskenazi, 2016; Wu et al., 2016), travel planners (Peng et al., 2017), restaurant/hotel bookers (Budzianowski et al., 2017), chat-bots (Serban et al., 2017; Li et al., 2016) and text-based game agents (Narasimhan, Kulkarni, and Barzilay, 2015; He et al., 2015; Zelinka, 2018; Yuan et al., 2018; Côté et al., 2018). RL is currently being applied in all of these domains, facing new challenges in function approximation and exploration due to the larger action space.

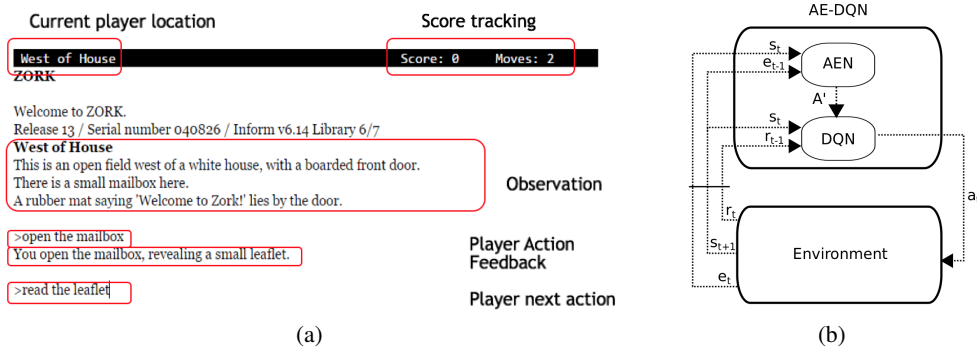

Figure 1: (*a*) **Zork interface.** The state in the game (observation) and the player actions are describe in natural language. (*b*) **The action elimination framework.** Upon taking action $a_t$, the agent observes a reward $r_t$, the next state $s_{t+1}$ and an elimination signal $e_t$. The agent uses this information to learn two function approximation deep networks: a DQN and an AEN. The AEN provides an admissible actions set $A'$ to the DQN, which uses this set to decide how to act and learn.

In this work, we propose a new approach for dealing with large action spaces that is based on *action elimination*; that is, restricting the available actions in each state to a subset of the most likely ones (Figure 1(b)). We propose a method that eliminates actions by utilizing an elimination signal; a specific form of an auxiliary reward (Jaderberg et al., 2016), which incorporates domain-specific knowledge in text games. Specifically, it provides the agent with immediate feedback regarding taken actions that are not optimal. In many domains, creating an elimination signal can be done using rule-based systems. For example, in parser-based text games, the parser gives feedback regarding irrelevant actions *after* the action is played (e.g., Player: "Climb the tree." Parser: "There are no trees to climb"). Given such signal, we can train a machine learning model to predict it and then use it to generalize to unseen states. Since the elimination signal provides immediate feedback, it is faster to learn which actions to eliminate (e.g., with a contextual bandit using the elimination signal) than to learn the optimal actions using only the reward (due to long term consequences). Therefore, we can design an algorithm that enjoys better performance by exploring invalid actions less frequently.

More specifically, we propose a system that learns an approximation of the Q-function and *concurrently learns to eliminate actions*. We focus on tasks where natural language characterizes both the states and the actions. In particular, the actions correspond to fixed length sentences defined over a finite dictionary (of words). In this case, the action space is of combinatorial size (in the length of the sentence and the size of the dictionary) and irrelevant actions must be eliminated to learn. We introduce a novel DRL approach with two DNNs, a DQN and an Action Elimination Network (AEN), both designed using a Convolutional Neural Network (CNN) that is suited to NLP tasks (Kim, 2014). Using the last layer activations of the AEN, we design a linear contextual bandit model that eliminates irrelevant actions with high probability, balancing exploration/exploitation, and allowing the DQN to explore and learn Q-values only for valid actions.

We tested our method in a text-based game called "Zork". This game takes place in a virtual world in which the player interacts with the world through a text-based interface (see Figure 1(a)). The player can type in any command, corresponding to the in-game action. Since the input is text-based, this yields more than a thousand possible actions in each state (e.g., "open door", "open mailbox" etc.). We demonstrate the agent's ability to advance in the game faster than the baseline agents by eliminating irrelevant actions.

## 2 Related Work

**Text-Based Games (TBG):** Video games, via interactive learning environments like the Arcade Learning Environment (ALE) (Bellemare et al., 2013), have been fundamental to the development of DRL algorithms. Before the ubiquitousness of graphical displays, TBG like Zork were popular in the adventure gaming and role-playing communities. TBG present complex, interactive simulations which use simple language to describe the state of the environment, as well as reporting the effects of player actions (See Figure 1(a)). Players interact with the environment through text commands that respect a predefined grammar, which must be discovered in each game.

TBG provide a testbed for research at the intersection of RL and NLP, presenting a broad spectrum of challenges for learning algorithms (Côté et al., 2018)[1]. In addition to language understanding, successful play generally requires long-term memory, planning, exploration (Yuan et al., 2018), affordance extraction (Fulda et al., 2017), and common sense. Text games also highlight major open challenges for RL: the action space (text) is combinatorial and compositional, while game states are partially observable since text is often ambiguous or under-specific. Also, TBG often introduce stochastic dynamics, which is currently missing in standard benchmarks (Machado et al., 2017). For example, in Zork, there is a random probability of a troll killing the player. A thief can appear (also randomly) in each room.

**Representations for TBG:** To learn control policies from high-dimensional complex data such as text, good word representations are necessary. Kim (2014) designed a shallow word-level CNN and demonstrated state-of-the-art results on a variety of classification tasks by using word embeddings. For classification tasks with millions of labeled data, random embeddings were shown to outperform state-of-the-art techniques (Zahavy et al., 2018). On smaller data sets, using *word2vec* (Mikolov et al., 2013) yields good performance (Kim, 2014).

Previous work on TBG used pre-trained embeddings directly for control (Kostka et al., 2017; Fulda et al., 2017). Other works combined pre-trained embeddings with neural networks. For example, He et al. (2015) proposed to use Bag Of Words features as an input to a neural network, learned separate embeddings for states and actions, and then computed the Q function from autocorrelations between these embeddings. Narasimhan et al. (2015) suggested to use a word level Long Short-Term Memory (Hochreiter and Schmidhuber, 1997) to learn a representation end-to-end, and Zelinka et al. (2018), combined these approaches.

**DRL with linear function approximation:** DRL methods such as the DQN have achieved state-of-the-art results in a variety of challenging, high-dimensional domains. This success is mainly attributed to the power of deep neural networks to learn rich domain representations for approximating the value function or policy (Mnih et al., 2015; Zahavy, Ben-Zrihem, and Mannor, 2016; Zrihem, Zahavy, and Mannor, 2016). Batch reinforcement learning methods with linear representations, on the other hand, are more stable and enjoy accurate uncertainty estimates. Yet, substantial feature engineering is necessary to achieve good results. A natural attempt at getting the best of both worlds is to learn a (linear) control policy on top of the representation of the last layer of a DNN. This approach was shown to refine the performance of DQNs (Levine et al., 2017) and improve exploration (Azizzadenesheli, Brunskill, and Anandkumar, 2018). Similarly, for contextual linear bandits, Riquelme et al. showed that a neuro-linear Thompson sampling approach outperformed deep (and linear) bandit algorithms in practice (Riquelme, Tucker, and Snoek, 2018).

**RL in Large Action Spaces:** Being able to reason in an environment with a large number of discrete actions is essential to bringing reinforcement learning to a larger class of problems. Most of the prior work concentrated on factorizing the action space into binary subspaces (Pazis and Parr, 2011; Dulac-Arnold et al., 2012; Lagoudakis and Parr, 2003). Other works proposed to embed the discrete actions into a continuous space, use a continuous-action policy gradient to find optimal actions in the continuous space, and finally, choose the nearest discrete action (Dulac-Arnold et al., 2015; Van Hasselt and Wiering, 2009). He et. al. (2015) extended DQN to unbounded (natural language) action spaces. His algorithm learns representations for the states and actions with two different DNNs and then models the Q values as an inner product between these representation vectors. While this approach can generalize to large action spaces, in practice, they only considered a small number of available actions (4) in each state.

Learning to eliminate actions was first mentioned by (Even-Dar, Mannor, and Mansour, 2003) who studied elimination in multi-armed bandits and tabular MDPs. They proposed to learn confidence intervals around the value function in each state and then use it to eliminate actions that are not optimal with high probability. Lipton et al. (2016a) studied a related problem where an agent wants to avoid catastrophic forgetting of dangerous states. They proposed to learn a classifier that detects hazardous states and then use it to shape the reward of a DQN agent. Fulda et al. (2017) studied affordances, the set of behaviors enabled by a situation, and presented a method for affordance extraction via inner products of pre-trained word embeddings.

# 3 Action Elimination

We now describe a learning algorithm for MDPs with an elimination signal. Our approach builds on the standard RL formulation (Sutton and Barto, 1998). At each time step $t$, the agent observes a state $s_t$ and chooses a discrete action $a_t \in \{1, .., |A|\}$. After executing the action, the agent obtains a reward $r_t(s_t, a_t)$ and observes the next state $s_{t+1}$ according to a transition kernel $P(s_{t+1}|s_t, a_t)$. The goal of the algorithm is to learn a policy $\pi(a|s)$ that maximizes the discounted cumulative return $V^\pi(s) = \mathbb{E}^\pi[\sum_{t=0}^\infty \gamma^t r(s_t, a_t)|s_0 = s]$ where $0 < \gamma < 1$ is the discount factor and $V$ is the value function. The optimal value function is given by $V^*(s) = \max_\pi V^\pi(s)$ and the optimal policy by $\pi^*(s) = \arg\max_\pi V^\pi(s)$. The Q-function $Q^\pi(s, a) = \mathbb{E}^\pi[\sum_{t=0}^\infty \gamma^t r(s_t, a_t)|s_0 = s, a_0 = a]$ corresponds to the value of taking action $a$ in state $s$ and continuing according to policy $\pi$. The optimal Q-function $Q^*(s, a) = Q^{\pi^*}(s, a)$ can be found using the Q-learning algorithm (Watkins and Dayan, 1992), and the optimal policy is given by $\pi^*(s) = \arg\max_a Q^*(s, a)$.

After executing an action, the agent also observes a binary elimination signal $e(s, a)$, which equals 1 if action $a$ may be eliminated in state $s$; that is, any optimal policy in state $s$ will never choose action $a$ (and 0 otherwise). The elimination signal can help the agent determine which actions not to take, thus aiding in mitigating the problem of large discrete action spaces. We start with the following definitions:

**Definition 1.** *Valid state-action pairs with respect to an elimination signal are state action pairs which the elimination process should not eliminate.*

As stated before, we assume that the set of valid state-action pairs contains all of the state-action pairs that are a part of some optimal policy, i.e., only strictly suboptimal state-actions can be invalid.

**Definition 2.** *Admissible state-action pairs with respect to an elimination algorithm are state action pairs which the elimination algorithm does not eliminate.*

In the following section, we present the main advantages of action elimination in MDPs with large action spaces. Afterward, we show that under the framework of linear contextual bandits (Chu et al., 2011), probability concentration results (Abbasi-Yadkori, Pal, and Szepesvari, 2011) can be adapted to guarantee that action elimination is correct in high probability. Finally, we prove that Q-learning coupled with action elimination converges.

## 3.1 Advantages in action elimination

Action elimination allows the agent to overcome some of the main difficulties in large action spaces, namely: Function Approximation and Sample Complexity.

*Function Approximation:* It is well known that errors in the Q-function estimates may cause the learning algorithm to converge to a suboptimal policy, a phenomenon that becomes more noticeable in environments with large action spaces (Thrun and Schwartz, 1993). Action elimination may mitigate this effect by taking the max operator only on valid actions, thus, reducing potential overestimation errors. Another advantage of action elimination is that the Q-estimates need only be accurate for valid actions. The gain is two-fold: first, there is no need to sample invalid actions for the function approximation to converge. Second, the function approximation can learn a simpler mapping (i.e., only the Q-values of the valid state-action pairs), and therefore may converge faster and to a better solution by ignoring errors from states that are not explored by the Q-learning policy (Hester et al., 2018).

*Sample Complexity*: The sample complexity of the MDP measures the number of steps, during learning, in which the policy is not $\epsilon$-optimal (Kakade and others, 2003). Assume that there are $A'$ actions that should be eliminated and are $\epsilon$-optimal, i.e., their value is at least $V^*(s) - \epsilon$. According to lower bounds by (Lattimore and Hutter, 2012), We need at least $\epsilon^{-2}(1 - \gamma)^{-3} \log 1/\delta$ samples per state-action pair to converge with probability $1 - \delta$. If, for example, the eliminated action returns no reward and doesn't change the state, the action gap is $\epsilon = (1 - \gamma)V^*(s)$, which translates to $V^*(s)^{-2}(1 - \gamma)^{-5} \log 1/\delta$ 'wasted' samples for learning each invalid state-action pair. For large $\gamma$, this can lead to a tremendous number of samples (e.g., for $\gamma = 0.99, \ (1 - \gamma)^{-5} = 10^{10}$). Practically, elimination algorithms can eliminate these actions substantially faster, and can, therefore, speed up the learning process approximately by $A/A'$ (such that learning is effectively performed on the valid state-action pairs).

Embedding the elimination signal into the MDP is not trivial. One option is to shape the original reward by adding an elimination penalty. That is, decreasing the rewards when selecting the wrong actions. Reward shaping, however, is tricky to tune, may slow the convergence of the function approximation, and is not sample efficient (irrelevant actions are explored). Another option is to design a policy that is optimized by interleaved policy gradient updates on the two signals, maximizing the reward and minimizing the elimination signal error. The main difficulty with this approach is that both models are strongly coupled, and each model affects the observations of the other model, such that the convergence of any of the models is not trivial.

Next, we present a method that decouples the elimination signal from the MDP by using contextual multi-armed bandits. The contextual bandit learns a mapping from states (represented by context vectors $x(s)$) to the elimination signal $e(s, a)$ that estimates which actions should be eliminated. We start by introducing theoretical results on linear contextual bandits, and most importantly, concentration bounds for contextual bandits that require almost no assumptions on the context distribution. We will later show that under this model we can decouple the action elimination from the learning process in the MDP, allowing us to learn using standard Q-learning while eliminating actions correctly.

### 3.2  Action elimination with contextual bandits

Let $x(s_t) \in \mathbb{R}^d$ be the feature representation of state $s_t$. We assume (realizability) that under this representation there exists a set of parameters $\theta_a^* \in \mathbb{R}^d$ such that the elimination signal in state $s_t$ is $e_t(s_t, a) = \theta_a^{*T} x(s_t) + \eta_t$, where $\|\theta_a^*\|_2 \leq S$. $\eta_t$ is an $R$-subgaussian random variable with zero mean that models additive noise to the elimination signal. When there is no noise in the elimination signal, then $R = 0$. Otherwise, as the elimination signal is bounded in $[0, 1]$, it holds that $R \leq 1$. We'll also relax our previous assumptions and allow the elimination signal to have values $0 \leq \mathbb{E}[e_t(s_t, a)] \leq \ell$ for any valid action and $u \leq \mathbb{E}[e_t(s_t, a)] \leq 1$ for any invalid action, with $\ell < u$. Next, we denote by $X_{t,a}$ ($E_{t,a}$) the matrix (vector) whose rows (elements) are the observed state representation vectors (elimination signals) in which action $a$ was chosen, up to time $t$. For example, the $i^{th}$ row in $X_{t,a}$ is the representation vector of the $i^{th}$ state on which the action $a$ was chosen. Denote the solution to the regularized linear regression $\|X_{t,a}\theta_{t,a} - E_{t,a}\|_2^2 + \lambda\|\theta_{t,a}\|_2^2$ (for some $\lambda > 0$) by $\hat{\theta}_{t,a} = \bar{V}_{t,a}^{-1} X_{t,a}^T E_{t,a}$ where $\bar{V}_{t,a} = \lambda I + X_{t,a}^T X_{t,a}$.

Similarly to Theorem 2 in (Abbasi-Yadkori, Pal, and Szepesvari, 2011)[2], for any state history and with probability of at least $1 - \delta$, it holds for all $t > 0$ that $\left| \hat{\theta}_{t,a}^T x(s_t) - \theta_a^{*T} x(s_t) \right| \leq \sqrt{\beta_t(\delta) x(s_t)^T \bar{V}_{t,a}^{-1} x(s_t)}$, where $\sqrt{\beta_t(\delta)} = R\sqrt{2\log(\det(\bar{V}_{t,a})^{1/2}\det(\lambda I)^{-1/2}/\delta)} + \lambda^{1/2}S$. If $\forall s, \|x(s)\|_2 \leq L$, then $\beta_t$ can be bounded by $\sqrt{\beta_t(\delta)} \leq R\sqrt{d\log\left(\frac{1+tL^2/\lambda}{\delta}\right)} + \lambda^{1/2}S$. Next, we define $\tilde{\delta} = \delta/k$ and bound this probability for all the actions, i.e., $\forall a, t > 0$

$$\Pr\left\{ \left| \hat{\theta}_{t-1,a}^T x(s_t) - \theta_{t-1,a}^{*T} x(s_t) \right| \leq \sqrt{\beta_t(\tilde{\delta}) x(s_t)^T \bar{V}_{t-1,a}^{-1} x(s_t)} \right\} \geq 1 - \delta \tag{1}$$

Recall that any valid action $a$ at state $s$ satisfies $\mathbb{E}[e_t(s, a)] = \theta_a^{*T} x(s_t) \leq \ell$. Thus, we can eliminate action $a$ at state $s_t$ if

$$\hat{\theta}_{t-1,a}^T x(s_t) - \sqrt{\beta_{t-1}(\tilde{\delta}) x(s_t)^T \bar{V}_{t-1,a}^{-1} x(s_t)} > \ell \tag{2}$$

This ensures that with probability $1 - \delta$ we never eliminate any valid action. We emphasize that only the expectation of the elimination signal is linear in the context. The expectation does not have to be binary (while the signal itself is). For example, in conversational agents, if a sentence is not understood by 90% of the humans who hear it, it is still desirable to avoid saying it. We also note that we assume $\ell$ is known, but in most practical cases, choosing $\ell \approx 0.5$ should suffice. In the current formulation, knowing $u$ is not necessary, though its value will affect the overall performance.

## 3.3 Concurrent Learning

We now show how the Q-learning and contextual bandit algorithms can learn simultaneously, resulting in the convergence of both algorithms, i.e., finding an optimal policy and a minimal valid action space. The challenge here is that each learning process affects the state-action distribution of the other. We first define Action Elimination Q-learning.

**Definition 3.** *Action Elimination Q-learning is a Q-learning algorithm which updates only admissible state-action pairs and chooses the best action in the next state from its admissible actions. We allow the base Q-learning algorithm to be any algorithm that converges to $Q^*$ with probability 1 after observing each state-action infinitely often.*

If the elimination is done based on the concentration bounds of the linear contextual bandits, we can ensure that Action Elimination Q-learning converges, as can be seen in Proposition 1 (See Appendix A for a full proof).

**Proposition 1.** *Assume that all state action pairs $(s, a)$ are visited infinitely often unless eliminated according to $\hat{\theta}_{t-1,a}^T x(s) - \sqrt{\beta_{t-1}(\tilde{\delta}) x(s)^T \bar{V}_{t-1,a}^{-1} x(s)} > \ell$. Then, with a probability of at least $1 - \delta$, action elimination Q-learning converges to the optimal Q-function for any valid state-action pairs. In addition, actions which should be eliminated are visited at most $T_{s,a}(t) \leq 4\beta_t/(u - \ell)^2 + 1$ times.*

Notice that when there is no noise in the elimination signal ($R = 0$), we correctly eliminate actions with probability 1, and invalid actions will be sampled a finite number of times. Otherwise, under very mild assumptions, invalid actions will be sampled a logarithmic number of times.

## 4 Method

Using raw features like *word2vec*, directly for control, results in exhaustive computations. Moreover, raw features are typically not realizable, i,.e., the assumption that $e_t(s_t, a) = \theta_a^{*T} x(s_t) + \eta_t$ does not hold. Thus, we propose learning a set of features $\phi(s_t)$ that are realizable, i.e., $e(s_t, a) = \theta_a^{*T} \phi(s_t)$, using neural networks (using the activations of the last layer as features). A practical challenge here is that the features must be fixed over time when used by the contextual bandit, while the activations change during optimization. We therefore follow a batch-updates framework (Levine et al., 2017; Riquelme, Tucker, and Snoek, 2018), where every few steps we learn a new contextual bandit model that uses the last layer activations of the AEN as features.

---

**Algorithm 1** deep Q-learning with action elimination

---

**Input:** $\epsilon, \beta, \ell, \lambda, C, L, N$
Initialize AEN and DQN with random weights $\omega, \theta$ respectively, and set target networks $Q^-, E^-$ with a copy of $\theta, \omega$
Define $\phi(s) \leftarrow \text{LastLayerActivations}(E(s))$
Initialize Replay Memory D to capacity N
**for** t = 1,2,..., **do**
    $a_t = \text{ACT}(s_t, Q, E^-, V^{-1}, \epsilon, \ell, \beta)$
    Execute action $a_t$ and observe $\{r_t, e_t, s_{t+1}\}$
    Store transition $\{s_t, a_t, r_t, e_t, s_{t+1}\}$ in D
    Sample transitions
$$\{s_j, a_j, r_j, e_j, s_{j+1}\}_{j=1}^m \in D$$
    $y_j = \text{Targets}\left(s_{j+1}, r_j, \gamma, Q^-, E^-, V^{-1}, \beta, \ell\right)$
    $\theta = \theta - \nabla_\theta \sum_j (y_j - Q(s_j, a_j; \theta))^2$
    $\omega = \omega - \nabla_\omega \sum_j (e_j - E(s_j, a_j; \omega))^2$
    If $(t \bmod C) = 0 : Q^- \leftarrow Q$
    If $(t \bmod L) = 0$ :
        $E^-, V^{-1} \leftarrow \text{AENUpdate}(E, \lambda, D)$
**end for**

**function** ACT$(s, Q, E, V^{-1}, \epsilon, \beta, \ell)$
    $A' \leftarrow \{a : E(s)_a - \sqrt{\beta \phi(s)^T V_a^{-1} \phi(s)} < \ell\}$
    With probability $\epsilon$, return Uniform$(A')$
    Otherwise, return $\arg\max_{a \in A'} Q(s, a)$
**end function**
**function** TARGETS$(s, r, \gamma, Q, E, V^{-1}, \beta, \ell)$
    **if** $s$ is a terminal state **then** return $r$ **end if**
    $A' \leftarrow \{a : E(s)_a - \sqrt{\beta \phi(s)^T V_a^{-1} \phi(s)} < \ell\}$
    return $(r + \gamma \max_{a \in A'} Q(s, a))$
**end function**
**function** AENUPDATE$(E^-, \lambda, D)$
    **for** $a \in A$ **do**
        $V_a^{-1} = \left(\sum_{j:a_j=a} \phi(s_j)\phi(s_j)^T + \lambda I\right)^{-1}$
        $b_a = \sum_{j:a_j=a} \phi(s_j)^T e_j$
        Set LastLayer$(E_a^-) \leftarrow V_a^{-1} b_a$
    **end for**
    return $E^-, V^{-1}$
**end function**

---

Our Algorithm presents a hybrid approach for DRL with Action Elimination (AE), by incorporating AE into the well-known DQN algorithm to yield our AE-DQN (Algorithm 1 and Figure 1(b)). AE-DQN trains two networks: a DQN denoted by $Q$ and an AEN denoted by $E$. The algorithm uses $E$, and creates a linear contextual bandit model from it every $L$ iterations with procedure **AENUpdate()**. This procedure uses the activations of the last hidden layer of $E$ as features, $\phi(s) \leftarrow$ LastLayerActivations$(E(s))$, which are then used to create a contextual linear bandit model ($V_a = \lambda I + \sum_{j:a_j=a} \phi(s_j)\phi(s_j)^T, b_a = \sum_{j:a_j=a} \phi(s_j)^T e_j$). AENUpdate() proceeds by solving this model, and plugging the solution into the target AEN (LastLayer$(E_a^-) \leftarrow V_a^{-1} b_a$). The contextual linear bandit model ($E^-, V$) is then used to eliminate actions (with high probability) via the **ACT()** and **Targets()** functions. ACT() follows an $\epsilon-$greedy mechanism on the admissible actions set $A' = \{a : E(s)_a - \sqrt{\beta \phi(s)^T V_a^{-1} \phi(s)} < \ell\}$. If it decides to exploit, then it selects the action with highest $Q$-value by taking an $\arg\max$ on $Q$-values among $A'$, and if it chooses to explore, then, it selects an action uniformly from $A'$. **Targets()** estimates the value function by taking $\max$ over $Q$-values only among admissible actions, hence, reducing function approximation errors.

**Architectures:** The agent uses an Experience Replay (Lin, 1992) to store information about states, transitions, actions, and rewards. In addition, our agent also stores the elimination signal, provided by the environment (Figure 1(b)). The architecture for both the AEN and DQN is an NLP CNN, based on (Kim, 2014). We represent the state as a sequence of words, composed of the game descriptor (Figure 1(a), "Observation") and the player's inventory. These are truncated or zero-padded (for simplicity) to a length of 50 (descriptor) + 15 (inventory) words and each word is embedded into continuous vectors using *word2vec* in $\mathbb{R}^{300}$. The features of the last four states are then concatenated together such that our final state representations $s$ are in $\mathbb{R}^{78,000}$. The AEN is trained to minimize the MSE loss, using the elimination signal as a label. We used $100$ ($500$ for DQN) convolutional filters, with three different 1D kernels of length (1,2,3) such that the last hidden layer size is 300. [3]

# 5   Experimental Results

**Grid World Domain:** We start with an evaluation of action elimination in a small grid world domain with 9 rooms, where we can carefully analyze the effect of action elimination. In this domain, the agent starts at the center of the grid and needs to navigate to its upper-left corner. On every step, the agent suffers a penalty of $(-1)$, with a terminal reward of 0. Prior to the game, the states are randomly divided into $K$ categories. The environment has $4K$ navigation actions, 4 for each category, each with a probability to move in a random direction. If the chosen action belongs to the same category as the state, the action is performed correctly with probability $p_c^T = 0.75$. Otherwise, it will be performed correctly with probability $p_c^F = 0.5$. If the action does not fit the state category, the elimination signal equals 1, and if the action and state belong to the same category, then it equals 0. An optimal policy only uses the navigation actions from the same type as the state, as the other actions are clearly suboptimal. We experimented with a vanilla Q-learning agent without action elimination and a tabular version of action elimination Q-learning. Our simulations show that action elimination dramatically improves the results in large action spaces. In addition, we observed that the gain from action elimination increases as the amount of categories grows, and as the grid size grows, since the elimination allows the agent to reach the goal earlier. We have also experimented with random elimination signal and other modifications in the domain. Due to space constraints, we refer the reader to the appendix for figures and visualization of the domain.

**Zork domain:** *"This is an open field west of a white house, with a boarded front door. There is a small mailbox here. A rubber mat saying 'Welcome to Zork!' lies by the door".* This is an excerpt from the opening scene provided to a player in "Zork I: The Great Underground Empire"; one of the first interactive fiction computer games, created by members of the MIT Dynamic Modeling Group in the late 70s. By exploring the world via interactive text-based dialogue, the players progress in the game. The world of Zork presents a rich environment with a large state and action space (Figure 2). Zork players describe their actions using natural language instructions. For example, in the opening excerpt, an action might be 'open the mailbox' (Figure 1(a)). Once the player describes his/her action, it is processed by a sophisticated natural language parser. Based on the parser's results, the game presents the outcome of the action. The ultimate goal of Zork is to collect the Twenty Treasures of

`https://github.com/TomZahavy/CB_AE_DQN`

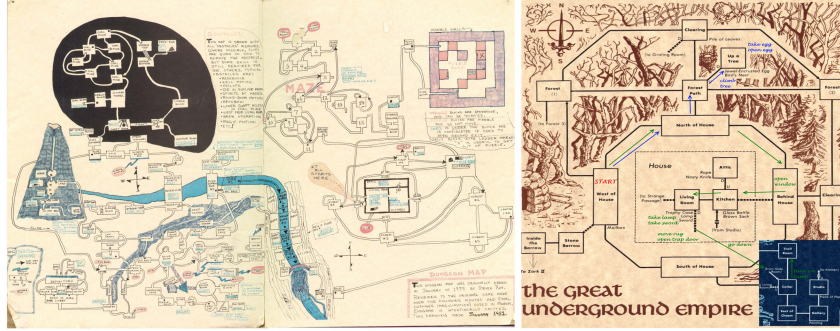

Figure 2: **Left:** the world of Zork. **Right:** subdomains of Zork; the Troll (green) and Egg (blue) Quests. Credit: S. Meretzky, The Strong National Museum of Play. Larger versions in Appendix B.

Zork and install them in the trophy case. Finding the treasures require solving a variety of puzzles such as navigation in complex mazes and intricate action sequences. During the game, the player is awarded points for performing deeds that bring him closer to the game's goal (e.g., solving puzzles). Placing all of the treasures into the trophy case generates a total score of 350 points for the player. Points that are generated from the game's scoring system are given to the agent as a reward. Zork presents multiple challenges to the player, like building plans to achieve long-term goals; dealing with random events like troll attacks; remembering implicit clues as well as learning the interactions between objects in the game and specific actions. The **elimination signal** in Zork is given by the Zork environment in two forms, a "wrong parse" flag, and text feedback (e.g. "you cannot take that"). We group these two signals into a single binary signal which we then provide to our learning algorithm. Before we started experimenting in the "Open Zork" domain, i.e., playing in Zork without any modifications to the domain, we evaluated the performance on two subdomains of Zork. These subdomains are inspired by the Zork plot and are referred to as the Egg Quest and the Troll Quest (Figure 2, right, and Appendix B). For these subdomains, we introduced an additional reward signal (in addition to the reward provided by the environment) to guide the agent towards solving specific tasks and make the results more visible. In addition, a reward of $-1$ is applied at every time step to encourage the agent to favor short paths. When solving "Open Zork" we only use the environment reward. The optimal time that it takes to solve each quest is 6 in-game timesteps for the Egg quest, 11 for the Troll quest and 350 for "Open Zork". The agent's goal in each subdomain is to maximize its cumulative reward. Each trajectory terminates upon completing the quest or after $T$ steps are taken. We set the discounted factor during training to $\gamma = 0.8$ but use $\gamma = 1$ during evaluation [4]. We used $\beta = 0.5, \ell = 0.6$ in all the experiments. The results are averaged over 5 random seeds, shown alongside error bars (std/3).

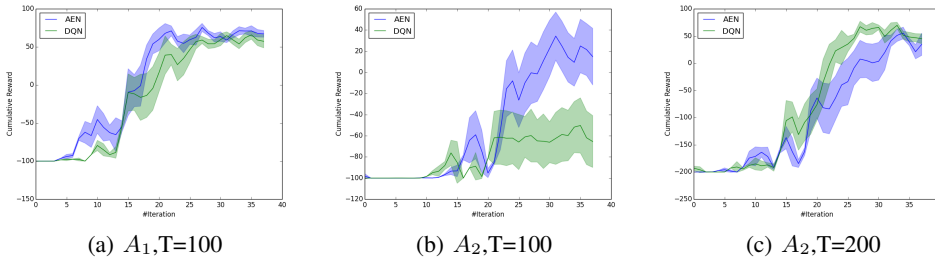

(a) $A_1$,T=100        (b) $A_2$,T=100        (c) $A_2$,T=200

Figure 3: Performance of agents in the egg quest.

**The Egg Quest:** In this quest, the agent's goal is to find and open the jewel-encrusted egg, hidden on a tree in the forest. The agent is awarded 100 points upon successful completion of this task. We experimented with the AE-DQN (blue) agent and a vanilla DQN agent (green) in this quest (Figure 3). The action set in this quest is composed of two subsets. A fixed subset of 9 actions that allow it to complete the Egg Quest like *navigate* (south, east etc.) *open* an item and *fight*; And a second subset consists of $N_{\text{Take}}$ *"take"* actions for possible objects in the game. The "take" actions correspond to taking a single object and include objects that need to be collected to complete quests, as well as

other irrelevant objects from the game dictionary. We used two versions of this action set, $A_1$ with $N_{\text{Take}} = 200$ and $A_2$ with $N_{\text{Take}} = 300$. **Robustness to hyperparameter tuning:** We can see that for $A_1$, with T=100, (Figure 3$a$), and for $A_2$, with T=200, (Figure 3$c$) *Both* agents solve the task well. However, for $A_2$, with T=100, (Figure 3$b$) the AE-DQN agent learns considerably faster, implying that action elimination is more robust to hyperparameters settings when there are many actions.

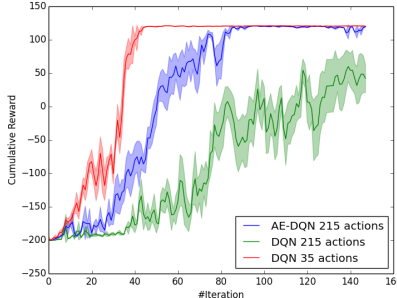

Figure 4: Results in the Troll Quest.

**The Troll Quest:** In this quest, the agent must find a way to enter the house, grab a lantern and light it, expose the hidden entrance to the underworld and then find the troll, awarding him 100 points. The Troll Quest presents a larger problem than the Egg Quest, but smaller than the full Zork domain; it is large enough to gain a useful understanding of our agents' performance. The AE-DQN (blue) and DQN (green) agents use a similar action set to $A_1$ with 200 take actions and 15 necessary actions (215 in total). For comparison, We also included an "optimal elimination" baseline (red) that consists of only 35 actions (15 essential, and 20 relevant take actions). We can see in Figure 5 that AE-DQN significantly outperforms DQN, achieving compatible performance to the "optimal elimination" baseline. In addition, we can see that the improvement of the AE-DQN over DQN is more significant in the Troll Quest than the Egg quest. This observation is consistent with our tabular experiments.

**"Open Zork":** Next, we evaluated our agent in the "Open Zork" domain (without hand-crafting reward and termination signals). To compare our results with previous work, we trained our agent for 1M steps: each trajectory terminates after $T = 200$ steps, and a total of 5000 trajectories were executed [5]. We used two action sets: $A_3$, the "Minimal Zork" action set, is the minimal set of actions (131) that is required to solve the game (comparable with the action set used by Kostka et al. (2017)). The actions are taken from a tutorial for solving the game. $A_4$, the "Open

Table 1: Experimental results in Zork

|  | $|A|$ | cumulative reward |
|---|---|---|
| Kostka et al. 2017 | $\approx 150$ | 13.5 |
| Ours, $A_3$ | 131 | **39** |
| Ours, $A_3$, 2M steps | 131 | **44** |
| Fulda et al. 2017 | $\approx 500$ | 8.8 |
| Ours, $A_4$ | 1227 | **16** |
| Ours, $A_4$, 2M steps | 1227 | **16** |

Zork" action set, includes 1227 actions (comparable with Fulda et al. (2017)). This set is created from action "templates", composed of {Verb, Object} tuples for all the verbs (19) and objects (62) in the game (e.g, open mailbox). In addition, we include a fixed set of 49 actions of varying length (but not of length 2) that are required to solve the game. Table 1 presents the average (over seeds) maximal (in each seed) reward obtained by our AE-DQN agent in this domain while using action sets $A_3$ and $A_4$, showing that our agent achieves state-of-the-art results, outperforming all previous work. In the appendix, we show the learning curves for both AE-DQN and DQN agents. Again, we can see that AE-DQN outperforms DQN, learning faster and achieving more reward.

# 6 Summary

In this work, we proposed the AE-DQN, a DRL approach for eliminating actions while performing Q-learning, for solving MDPs with large state and action spaces. We tested our approach on the text-based game Zork, showing that by eliminating actions the size of the action space is reduced, exploration is more effective, and learning is improved. We provided theoretical guarantees on the convergence of our approach using linear contextual bandits. In future work, we plan to investigate more sophisticated architectures, as well as learning shared representations for elimination and control which may boost performance on both tasks. In addition, we aim to investigate other mechanisms for action elimination, e.g., eliminating actions that result from low Q-values (Even-Dar, Mannor, and Mansour, 2003). Another direction is to generate elimination signals in real-world domains. This can be done by designing a rule-based system for actions that should be eliminated, and then, training an AEN to generalize these rules for states that were not included in these rules. Finally, elimination signals may be provided implicitly, e.g., by human demonstrations of actions that should not be taken.

## Footnotes

[1]See also The CIG Competition for General Text-Based Adventure Game-Playing Agents

[2]Our theoretical analysis builds on results from (Abbasi-Yadkori, Pal, and Szepesvari, 2011), which can be extended to include Generalized Linear Models (GLMs). We focus on linear contextual bandits as they enjoy easier implementation and tighter confidence intervals in comparison to GLMs. We will later combine the bandit with feature approximation, which will approximately allow the realizability assumption even for linear bandits.

[3]Our code, the Zork domain, and the implementation of the elimination signal can be found at:

[4]We adopted a common evaluation scheme that is used in the ALE. During learning and training we use $\gamma < 1$ but evaluation is performed with $\gamma = 1$. Intuitively, during learning, choosing $\gamma < 1$ helps to learn, while during evaluation, the sum of cumulative returns ($\gamma = 1$) is more interpretable (the score in the game).

[5]The same amount of steps that were used in previous work on Zork (Fulda et al., 2017; Kostka et al., 2017). For completeness, we also report results for AE-DQN with 2M steps, where learning seemed to converge.

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
