[Supplementary Material · supplementary.pdf]

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

*Proof.* We start by proving the convergence of the algorithm and then prove the bound on the number of visits of invalid actions.

Denote the MDP as $M$. According to Equation 1, with probability of at least $1 - \delta$, elimination by Equation 2 never eliminates a valid action, and thus all of these actions are visited infinitely often. If all of the state-action pairs are visited infinitely often even after the elimination, the Q-learning will converge at all state-action pairs. Otherwise, there are some invalid actions, which are strictly suboptimal, and are visited a finite number of times. In this case, there exists some time $T < \infty$ such that all of these actions are never played for any $t > T$. Define a new MDP $\tilde{M}$, as $M$ without any of the eliminated actions. As these actions are strictly suboptimal, the value of $\tilde{M}$ will be identical to the value of $M$ in all states, and so are the Q-values for any action that survived the elimination. Furthermore, $\tilde{M}$ contains all of the valid states, and their Q-values will be identical those of $M$, as they only depend on the reward in the valid state-action pairs and the value in the next state, both which exist in $\tilde{M}$. For any $t > T$, $M$ is equivalent to $\tilde{M}$, and all of its state-actions are visited infinitely often. Therefore, the Q-function will converge to the optimal Q-function with probability 1 in all of $\tilde{M}$'s state-action pairs. Specifically, it will converge in all of valid state-action pairs $(s, a)$, which concludes the first part of the proof.

We'll now prove the sample complexity of any invalid actions. First, note that the confidence bound is strongly related to the number of visits in a state-action pair:

$$
\begin{aligned}
x(s_t)^T \bar{V}_{t-1,a}^{-1} x(s_t) &= x(s_t)^T \left\{ \lambda I + T_{s,a}(t-1)x(s_t)x(s_t)^T + \sum_{s' \neq s_t} T_{s',a}(t-1)x(s')x(s')^T \right\}^{-1} x(s_t) \\
&\overset{(1)}{\le} x(s_t)^T \left\{ \lambda I + T_{s,a}(t-1)x(s_t)x(s_t)^T \right\}^{-1} x(s_t) \\
&\overset{(2)}{=} \frac{\|x(s_t)\|^2}{\lambda} - \frac{T_{s,a}(t-1)\frac{\|x(s_t)\|^4}{\lambda^2}}{1 + T_{s,a}(t-1)\frac{\|x(s_t)\|^2}{\lambda}} \\
&= \frac{\|x(s_t)\|^2}{\lambda + T_{s,a}(t-1)\|x(s_t)\|^2} \le \frac{1}{T_{s,a}(t-1)}
\end{aligned}
\tag{3}
$$

$(1)$ is correct due to the fact that for any positive definite $A$ and positive semidefinite $B$, the difference $A^{-1} - (A + B)^{-1}$ is positive semidefinite. $(2)$ is correct due to the Sherman–Morrison formula (Bartlett, 1951). We note that this bound is not tight because it does not use the correlations between different contexts. In fact, the same bound can be probably achieved by placing a multi-armed bandit algorithm in each state. Deriving a tighter bound that utilizes the correlation between contexts is hard, as it is possible to observe a state that its context is not correlated with other states' contexts. Nevertheless, the confidence bounds for contextual bandits can be used in the non-tabular case, in contrast to a MAB formulation.

This implies that a satisfactory condition for correct elimination is

$$
\begin{aligned}
x(s_t)^T \hat{\theta}_{t-1,a} &- \sqrt{\beta_{t-1}(\tilde{\delta})x(s_t)^T \bar{V}_{t-1,a}^{-1} x(s_t)} \\
&\overset{(1)}{\ge} u - 2\sqrt{\beta_{t-1}(\tilde{\delta})x(s_t)^T \bar{V}_{t-1,a}^{-1} x(s_t)} \overset{(2)}{\ge} u - 2\sqrt{\frac{\beta_{t-1}(\tilde{\delta})}{T_{s,a}(t-1)}} > \ell
\end{aligned}
$$

where (1) is correct due to Equation 2 with $\mathbb{E}[e(s_t, a)] = \theta_a^{*T} x(s_t) \geq u$, with probability $1 - \delta$, and (2) is correct due to Equation 3. Therefore, if $T_{s,a}(t) \geq 4 \frac{\beta_t}{(u-\ell)^2}$ then action $a$ in state $s$ is correctly eliminated. We emphasize that the bound does not depend on the algorithm that chooses state-actions, except for the dependency of $\beta_t$, through $\bar{V}_{t,a}$, in the history. Using the fact that $\beta_t$ is monotonically increasing with $t$, with probability $1 - \delta$, all of the invalid actions are sampled no more than

$$T_{s,a}(t) \leq \sum_{\tau=1}^{t} \mathbb{1}\left\{T_{s,a}(\tau) \leq 4 \frac{\beta_\tau}{(u-\ell)^2}\right\}$$

$$\leq \sum_{\tau=1}^{t} \mathbb{1}\left\{T_{s,a}(\tau) \leq 4 \frac{\beta_t}{(u-\ell)^2}\right\} \leq 4 \frac{\beta_t}{(u-\ell)^2} + 1$$

If the sub-gaussianity parameter is $R = 0$, we have $\beta_t = \lambda S^2 < \infty$, and therefore an arm will be sampled at most a finite number of times $T_0 = 4 \frac{\lambda S^2}{(u-\ell)^2} + 1 < \infty$. Otherwise, if the state representations are bounded, i.e. $\forall s, \|x(s)\|_2 \leq L$, then, using the simpler form of $\beta_t$, the bound can be written as $\lim_{t\to\infty} \frac{T_{s,a}(t)}{\log\left(\frac{t}{\delta}\right)} \leq \frac{4R^2 d}{(u-\ell)^2}$ , which means an invalid action is sampled a logarithmic number of times.

□

## Appendix B    Grid world simulations

In this Section, we experimented with action elimination in a grid world domain with a tabular Q-learning algorithm. We start with the following default configuration (Figure 5). The grid size is 30x30, the number of state categories is $K = 10$, and the maximal episode length is $T = 150$. If the chosen action is from the same category as the current state, it is performed correctly in probability $p_c^T = 0.75$, and if the state and action types are different, the probability is $p_c^F = 0.5$. We also study the effect of the domain's parameters on the performance of action elimination, by changing these parameters one at a time.

Figure 5: 30x30 Grid World - the agent starts at the center (green) and needs to navigate to the upper-left corner (blue) while avoiding walls (yellow).

On each of the simulations, the results were filtered by a moving average filter of length 200, which is needed due to the stochastic nature of the domain. The results are averaged over 5 random seeds, shown alongside error bars (std/3).

Since the problem is tabular, we use confidence intervals in the spirit of UCT (Kocsis and Szepesvári, 2006) - denote the empirical mean of the elimination signal by $\bar{e}(s, a)$ and the number of visits in a state-action pair by $N(s, a)$. An action will be eliminated if $\bar{e}(s, a) - \sqrt{\frac{2 \sum_a N(s,a)}{N(s,a)}} > \ell \triangleq 0.5$. The Q-function was initially set to 0, the learning rates were chosen according to (Even-Dar and Mansour, 2003), and we set $\gamma = 1$.

We start with comparison between vanilla Q-learning without action elimination (green) and a tabular version of the action elimination Q-learning (blue) (Figure 6). We also include an "optimal elimination" baseline, i.e., a Q-learning agent with one category (red), i.e., only 4 basic navigation actions, which forms an upper bound on performance with multiple categories. In Figure 6(a),6(c), the episode length is $T = 150$, and in Figure 6(b) it is $T = 300$, to allow sufficient exploration for the vanilla Q-Learning.

(a) 30x30,K=1,10             (b) 30x30,K=1,25             (c) 20x20,K=1,10

Figure 6: Performance of agents in grid world.

We can see that action elimination significantly improves the Q-learning algorithm (blue) over the baseline (green). In Figure 6(b), we increased the number of state categories. We can see that in this case, the action elimination dramatically improves in comparison to the vanilla algorithms, since there are more invalid actions (compare Figure 6(a) with Figure 6(b)). Figure 7(a) present the simulation results with a 40x40 grid world. When compared to Figures 6(a),6(c), with grid of 30x30 and 20x20, respectively, we can conclude that action elimination becomes more effective as the grid size increases. Intuitively, it is relatively easy to reach the goal state on small grids, even if the action space is large, and therefore even random exploration will bring the agent to the goal quickly. From the moment the agent reaches the goal, its policy will be biased towards the goal's direction, and it becomes easier to distinguish between valid and invalid actions.

Next, we make a small modification in the domain and consider a random elimination signal, i.e., if the action does not fit the state category, an elimination signal will be equal 1 in probability $p_e^F$. If the action and state belong to the same category, the probability is $p_e^T$. Specifically, we let $p_e^F$ change between 1 to 0.6 in invalid actions, and $p_e^T$ between 0 to 0.4 in valid actions. Inspecting Figure 7(b), we observe that only when the elimination signal is almost completely random, the action elimination algorithm does not present superior performance.

Finally, Figure 7(c) present a scenario where valid actions has almost no randomness ($p_c^T = 0.9$), while invalid actions are almost completely random ($p_c^F = 0.1$). Thus, it is easier to identify the invalid actions, and specifically, understand that these actions are suboptimal. The results indeed show that while action elimination Q-learning converges faster than the vanilla Q-learning, the difference between the convergence rates is smaller.

(a) 40x40,K=1,10, T=300    (b) 30x30,different elimination prob-  (c) 30x30, different random step
                            ability                                probability

Figure 7: Performance of agents in grid world.

In summary, the tabular simulations showed a significant improvement due to action elimination, especially when the action space is large, the optimal and suboptimal actions are hard to distinct, and the horizon required to reach the goal is large.

# Appendix C    Additional figures

(a) $A_3$

(b) $A_4$

Figure 8: Results in "Open Zork".

# Appendix D  Maps of Zork

Figure 9: The world of Zork

Figure 10: Subdomains of Zork; the Troll (green) and Egg (blue) Quests. Credit: S. Meretzky, The Strong National Museum of Play.