[Reviews · NeurIPS 2018]

Reviewer 1



This paper addresses the challenge of an environment with discrete, but large number, of actions, by eliminating the actions that are never taken in a particular state. To do so, the paper proposes AE-DQN which augments DQN with contextual multi-armed bandit to identify actions that should be eliminated. Evaluation conducted on a text-based game, Zork, shows promising results, as AE-DQN outperforms baseline DQN on several examples. This idea of eliminating actions which are never taken in a given state is a sound on. The paper is clear and well written. On the flip side, I have several questions about the evaluation. First, the evaluation is pretty "thin", as it considers only three Zork tasks. Furthermore, in only one task (i.e. the troll quest) AE-DNQ achieves high reward and significantly outperforms the baseline, DQN. For another task, egg quest, both AE-DQN and DQN achieve similar rewards (though AE-DQN converges faster). Finally, for the hardest task, Open Zork, while AE-DQN performs better than previous results, it doesn't really solve the task achieving a reward no larger than 39 (out of 100). It would be great to add more tasks. Also, besides Zork it would be nice to consider other applications. Second, there is little discussion about the parameters used in various experiments. For instance, for the simpler tasks I believe you are using T = 100 steps, while for the more complex task, Open Zork, you are using T = 500 steps. You should provide more context. How well does your solution perform in the case of Open Zork for T=100? Also, how does the baseline perform for the simpler tasks if you use T = 500? Why doesn't DQN solve the test for the larger action set, i.e., bottom plot in Figure 5(a). I understand that it might take later to converge, but it should eventually converge. It would be nice to run the the experiment longer. Similarly, can you run the experiment in Figure 5(b) longer? It doesn't seem that the ACT + * solutions have achieved the plateau. During training you are using a discounted factor of 0.8 but, while for learning you are using a discounted factor of 1. Why?

Reviewer 2



In this paper, the authors propose AE-DQN, a deep RL algorithm for large action space tasks where at each time step, an addition signal on action elimination is also provided. AE-DQN learns an action elimination model which eliminates a part of action space and enhance the sample complexity of exploration. The paper is well written (can also be improved) and the importance of the problem is well motivated. The theoretical analysis is insightful and motivated. In order to make the paper more clear, I would suggest the author use Linear contextual bandit terminology instead of contextual bandit since they use linear pay-off framework. The paper motivates that the elimination signal is binary, therefore, I would suggest to the author to instead of providing theoretical insight using linear contextual bandit, please provide GLM style analysis where the output is actually a logistic. I believe expanding the current analysis to GLM is straightforward. If it is not, the authors can ignore the GLM part, but address more on the binary side of the motivation Regarding action elimination incorporated as negative penalties, the authors mention that adding negative reward might couple the MDP and elimination, therefore not desirable. I would recommend that the authors do an experimental comparison on it. For example, using Levin et al framework, but instead of e-greedy exploration on the best n_max actions, also the prior (lambda in the ridge regression), properly chosen such that the actions with low samples stay small.

Reviewer 3



This paper addresses the issue of having a very large action space when solving a Reinforcement Learning (RL) problem. The authors propose the Action-Elimination Deep Q-Network (AE-DQN) which is a model that eliminates sub-optimal actions. The training process requires an external supervised signal to learn which actions should be eliminated given the current state. The authors validate their approach by conducting experiments on text-based games where the number of discrete actions is over a thousand. They show a considerable speedup and added robustness over vanilla DQN. More precisely, in the model being proposed, the elimination signal is not used to simply shape the environment reward. Instead, it is decoupled from the MDP by using contextual multi-armed bandits. An Action Elimination Network learns a mapping from states to elimination signal, and given a state provides the subset of actions that should be used by the DQN (trained using standard Q-Learning). In addition, the authors show that their Action Elimination Q-Learning converges to find an optimal policy and a minimal valid action space in the contextual bandits setting. Main contributions: 1) Action-Elimination Deep Q-Network (AE-DQN) which is a model that eliminates sub-optimal actions in very large actions space using an external action elimination signal. 2) Convergence proof of the proposed Action Elimination Q-learning. 3) State of the art results on the text-based game Zork1. Pros: - A technique that deals with large action space is of great interest to the community; - Text-based games offer interesting challenges for RL algorithms (stochastic dynamics, delayed reward, long-term memory, language understanding) and have been little explored. - Overall, the paper is well written, and the authors communicate clearly the motivation and main ideas of the proposed method. Cons: - Extracting an elimination signal from most environments is not trivial. To be fair, the authors do address this concern in their conclusion. - It is unclear how the elimination signal is obtained for Zork1. Usually, in text-based games, that signal comes in the form of text which would need to be understood first (maybe some sort of sentiment analysis model?). - It is unclear to me what is the target of the NLP CNN classification. How does one determine the actions' relevance in each state? Is it by using the elimination signal? Minor issues: - I was not able to open the 'run_cpu' file in the anonymized code repo. From footnote #2, it seems the implementation of the elimination signal for Zork1 can be found there. I would still like to see it detailed in the paper (or at least in the Appendix). - The definition of the three action spaces is hard to follow. Maybe a table would be clearer if space allows it. - Figure 5(b)'s caption mentions 209 actions whereas in the text (Section 5.2) it says 215 actions. - It is unclear if the step penalty was used for the Open Zork domain? I would assume it isn't to make the results comparable to previous work. I would briefly mention it. EDIT: In light of the authors' answers, I'm raising my rating.